# Sport Injuries among Amateur Women and Young Intermediate Level Female Handball Players: A Preliminary Investigation

**DOI:** 10.3390/medicina57060565

**Published:** 2021-06-02

**Authors:** Miguel A. Sanchez-Lastra, Pedro Vicente Vila, Arón Díaz Ledo, Carlos Ayán

**Affiliations:** 1Departamento de Didácticas Especiais, Universidade de Vigo, 36310 Vigo, Spain; pvicente@uvigo.es (P.V.V.); arondiazledo8@gmail.com (A.D.L.); cayan@uvigo.es (C.A.); 2Well-Move Research Group, SERGAS-UVIGO Departamento de Didácticas Especiales, Facultad de Ciencias de la Educación y del Deporte, Universidad de Vigo, 36310 Vigo, Spain

**Keywords:** female, sports medicine, wounds and injuries, handball, injury prevention, health

## Abstract

*Background and objectives*: Handball is a popular sport among women whose practice can lead to marked health benefits but could also show a high injury risk. There is a lack of research on intermediate level female players. We aimed to identify the prevalence of sport injuries in amateur and young intermediate level female handball players and the potential influence of the sport category. *Materials and Methods*: A group of cadets, juvenile and senior female players of three handball clubs participating in the Spanish regional league were followed throughout the 2018–2019 season. Information on injuries and exposure was collected via questionnaires. *Results*: Out of 114 players (34 seniors, 33 juvenile and 47 cadets), 77 of them sustained at least one injury. Most of the injuries were either moderate or severe, regardless of the category. A total of 7.93, 5.93 and 4.16 injuries per 1000 h of exposure were registered in the senior, juvenile and cadet categories respectively. The risk of sustaining an injury was 2.14 times higher for senior players Confidence Interval (CI 95%: 1.51–3.03) and 1.92 higher for juvenile players (CI 95%: 1.32–2.78) when compared with cadets. *Conclusions*: Senior and young female handball athletes playing at an amateur and intermediate level, are exposed to a substantial risk of sustaining a moderate or severe injury. The players’ category does not seem to have a great impact on the injury rate and on the characteristics of the sport injuries registered.

## 1. Introduction

Promoting the practice of a team sport is considered an effective way to reduce sedentary behavior and improve the healthy lifestyle of adolescents [1]. This strategy is specially needed among girls, since they seem to be more sedentary and more prone to quit team sports than boys [2]. However, practicing sports carries an inherent risk of injury, which for young people may have both immediate and long-term consequences [3]. In this study, a sport injury is understood as any accident or physical impairment occurring during a practice or game that forced the player to be inactive (inability to participate in practices or games) for at least one day [4]. It has been argued that one of the psychological mediating factors in the occurrence of injuries is an elevated stress response, especially in regards to increased muscle tension, narrowing of the visual field and increased distractibility [5,6]. It is plausible to think that younger populations, which are exposed to both familiar and academic stressors could exacerbate the influence of these psychological factors. Thus, injury prevention studies focused on team sports, and specially in those performed by adolescent females, are needed.

In this scenario, team handball emerges as an interesting field of research, for being a popular sport among women whose practice can also lead to marked health benefits [7]. In Spain, children are federated and compete in handball from a very early age. Generally, they train two or three times a week and compete on weekends. Most of the children who start playing handball at an early age end up playing in senior teams of their clubs at an amateur level, because of the acquired habit of playing sports. However, it should be noted that handball is a contact sport that has shown a high injury risk in both senior and youth competitors [4], with serious injuries observed specially in young female players [4]. Consequently, it is vital to identify and describe the incidence and severity of the injuries of female handball players, as a first step towards developing effective injury-prevention strategies for this population.

Although several studies related to this topic have been published so far, most of them have been focused on elite or sub-elite women [8,9,10] and adolescent [11,12,13] players, while research carried out at amateur or intermediate competitive levels is less prominent [4,14]. Therefore, an update on this topic is welcomed. Similarly, there is a lack of research providing information further to the influence of the sport category (i.e., cadet, juvenile, senior) on the extent of the injury. This is an important fact to consider, since early identification of risk factors are needed for primary prevention of the subsequent sport injuries that might occur later on. Studying these aspects would increase the body of knowledge related to female handball players injury pattern, and therefore, it would allow a safer sport practice.

Under these circumstances, this research has a double fold objective. First, it aims at providing information regarding the injury characteristics observed in amateur senior and young intermediate level female handball players. A second objective is to identify the nature, extent, and severity of the observed injuries across different sport categories.

## 2. Materials and Methods

### 2.1. Study Design

Observational prospective cohort study [15] with Spanish female handball players conducted throughout the 2018–2019 season.

### 2.2. Participants

Female players enrolled in three handball clubs participating in the Spanish regional league were asked to take part in the investigation. For each club, a cadet (14–15 years), a juvenile (16–17 years) and a senior (>18 years) team were selected for the research. A total 114 players (seniors *n* = 34; juvenile *n* = 33 and cadets *n* = 47) took part in the study. None of them played on higher ranking during the season. A total of 57 players were receiving physical education lessons at high school. The Study conformed to the recommendations of the Declaration of Helsinki. Its design was reviewed and approved by the Ethics Committee of the Faculty of Education and Sports Science (University of Vigo) Reference Code: 11012, approved date: 18 January 2021.

### 2.3. Data Collection

Information on injuries and exposure was collected using a modified version of a questionnaire previously used for this purpose on young female basketball players [16]. The questionnaire was first adapted to handball and then e-mailed to the team coaches at the beginning of the pre-season and they were instructed to fill it in for each injury sustained by a player from the beginning of the pre-season to the end of the season. This procedure has been regarded as the best method to register injuries in youth team handball [4]. The questionnaire included items related to (a) number, type and anatomical site of injury; (b) type of playing surface regularly used for practice/competition; (c) background of injuries and injury recurrence, and (d), the cadet and juvenile players were asked about the number of days they were inactive due to injury and its self-perceived impact on academic performance (i.e., missed classes/educational activities). Information related to age, weight, height, date of injury, time spent per week on practice, time spent on games, field position, and how the injury occurred was also collected. 

According to the time absence from practice, injuries were classified as slight (able to practice handball the next day), minor (1–7 days), moderate (8–21 days) and major (>21 days) [4].

### 2.4. Statistical Analysis

The distribution of the variables in the sample was analyzed by visual inspection of histograms and the Shapiro-Wilk test [17]. In the case of non-normally distributed variables, Levene’s test was applied to check the homogeneity of variance. Descriptive statistics were used to show the characteristics of female handball players and injuries. The injury rate of handball female players was calculated multiplying the average length of training sessions/matches and the total number of matches of the season by the number of training sessions a week, considering simultaneous participation of fourteen female players in training sessions or seven female players in competition matches. The result is divided by the number of injuries and after that, multiplied by a thousand [18,19].

The data were analyzed using Chi contrast—Pearson square for the two-variant analysis between the dependent variable, incidence of injury, and the independent variables player position and time of absence, the level of significance is established in statistical terms at a *p*-value ≤ 0.005.

Odds Ratio (OR) was used to set an estimated value with confidence interval for the relationships between two variant –analysis determining the likelihood of an event happening or not. In our case, we compared whether cadets had a higher possibility of sustaining an injury than senior (*n* cadets plus senior = 81) or than juvenile (*n* cadets plus juvenile = 80) players. After this, juvenile and senior players were compared against each other (*n* = 69).

The analysis of the relationship between the category of the participants and the incidence of injury was calculated using OR and a 95% confidence interval (CI 95%). All analyses were performed in IBM SPSS Statistics 20.0 (Armonk, NY, USA: IBM Corp) for Windows.

## 3. Results

A total of 101 injuries were registered, 60 (59.41%) and 41 (40.59%) of them were sustained while compiting or while training respectively. The players characteristicas, main variables assessed and total of injuries registered are shown in Table 1. All variables showed a normal distribution except for age (*p* < 0.001), which showed no signficant differences for the variances across the study groups. Out of the 114 players who took part in the study, 77 of them (seniors *n* = 31 out of 34; juvenile *n* = 26 out of 33 and cadets *n* = 20 out 47) sustained at least one injury. These results indicated the existence of a mean injury rate of 1.26, 0.95 and 0.55 per player/season respectively (data not shown in the table). The majority of the injuries registered in each category were either moderate or severe, while around 28% of the injuries found in the senior category were classified as recurrent. No influence of the category on the risk of sustaining a moderate or severe injury was observed. When analysing injury distribution according to the playing positions, the highest percentages were found among back and wing players (25–35%). No significant associations were observed between the categories analysed and the injury distribution taking into account the players’ position. None of the variabless assessed seemed to have an impact on the registered injuries. The average time absent from sport practice was around three weeks, and players reported missing an average of 5 physical education lessons due to handball injuries throughout the season.

The characteristics of the total injuries registered are shown in Table 2. Regarding the type of injuries, it was found that traumatic injuries (69 out of 101) were almost twice as frequent as overuse ones (32 out 101) (68.3% vs. 38.6%), while contact injuries were almost equally distributed among the categories, overall. When stratifying the total of injuries registered by each category, it was found that musculoskeletal, sprains and other kind of injuries altogether, were the most frequently types observed in cadet (34.6%), junior (34.3%) and senior (32.5%), respectively. Regarding the type of action that led to an injury, it was observed that a high percentage of injuries were sustained while performing changes of direction and playing on cement/parquet flooring surfaces. A similar number of injuries were observed throughout the season phases. No significant differences among the three categories were found regarding the aforementioned variables.

Table 3 depicts the anatomical location of the registered injuries, overall and by sport category. Lower and upper-limb injuries were found mostly in cadet (18 out 26 registered injuries; 69.2%) and senior (26 out of 43 registered injuries; 60.4%) categories, respectively. The ankle and the knee, and the wrist-hand were the lower and upper-body anatomical locations most frequently hurt in almost all the categories. The risk ratio analysis indicated that senior players were 2.51 times more likely to sustain an upper-limb injury compared to cadet players (CI 95%: 1.50–5.27). 

The amount of hours spent in training and in competition in each category is shown in Table 4. 

A total of 7.93, 5.93 and 4.16 injuries per 1000 h of exposure were registered in the senior, juvenile and cadet categories respectively. Significant differences were not found among the three categories. According to the OR analysis, the risk of sustaining an injury was 2.14 times higher for senior players (CI 95%: 1.51–3.03) and 1.92 higher for juvenile players (CI 95%: 1.32–2.78) when compared with cadets (Table 5). No significant differences regarding injury risk were observed between senior and juvenile players.

## 4. Discussion

This study provides information regarding the characteristics of sport injuries registered on senior, juvenile and cadet amateur handball female players. The influence of the category on the characteristics and the chance for injury were also analyzed. The obtained data could be helpful for understanding the causes of injuries and implementing programs aimed at reducing their frequency or severity in this group of athletes. 

As far as we know, this is the first prospective study focused on amateur players from three different categories during one season. The discussion of the obtained findings is limited by this fact, since the existent research on female players is generally focused on elite, or elite and amateur participants and data is not reported separately, while some investigations have focused only on a specific type of injury [20]. In addition, we combined matches and training hours when analyzing the injury incidence, a procedure that has not always been used so far. For instance, Asai et al. [21], reported an injury incidence of 20.1 match injuries per 1000 h in female cadet players who took part in a one-month national tournament. Similarly, Wedderkopp et al. [14], found an injury incidence 40.7/1000 h of game, in a group of elite and amateur female juvenile players followed throughout a whole season. Regarding the injury rate per 1000 h of total exposure (training and competition), Moller et al. [8] found a higher injury incidence for cadets (6.8) but lower for juvenile (4.7) elite players after 31-weeks of competition. Other authors have reported even lower values (1.17/1000 h of practice) in young female players of different levels (elite, intermediate and recreational) [22]. In all of these investigations, injuries were registered trough questionnaires specifically designed by the authors, as it was the case of our research.

In comparison with the injury incidence observed in young amateur athletes practising other contact sports such as soccer or rugby (64.5 and 55.9/1000 h of exposure, respectively), ref [23] our results indicate that amateur handball is a less harmful practice. Nevertheless, taking into account the injury rate reported by Moller et al. [8] and Wedderkopp et al. [22], it seems that amateur young players who play at a regional level show a considerable risk of sustaining an injury. 

Regarding the injury incidence observed in the senior category, we obtained higher values, even when compared with elite and subelite female players followed during a season [10,24]. It should be taken into account that in the aforementioned investigations, the sample was made up of athletes who trained many more hours than our players. The number of injuries that occur during matches is usually higher than the number of injuries occurring during training sessions [20] and, as such, differences in the ratio of training and match hours may bias calculations of the overall incidence of injury [23]. Nevertheless, our data indicate the existence of a considerable injury risk for amateur senior female players. This can be assumed after observing that seniors had a lower training and competition load and still they showed a high number of players injured.

Injury incidence was higher in senior than in the other two categories. This finding could be a result of differences in training loads, season organization or maturation level. However, this is mere speculation since we did not register these variables. We have found only one investigation in which data regarding injury incidence in senior, juvenile and cadet female players was provided [8]. Although in this study, it was indicated that injury incidence falls with age, as in our case, no information regarding the existence of significant differences between the players’ categories was reported. In this regard, Achenbach, Loose et al. [25], found that senior players showed a more significant overall injury incidence than U17 players, although the research was focused on beach handball. Differences in both sport modalities, prevent further discussion. We also observed that there were no significant differences between juvenile and cadet players regarding injury incidence, a finding that it is in line with the results obtained by Achenbach, Krustch et al. [26]. 

Regarding risk ratio, we found that senior and juvenile players had a significant higher risk of sustaining and injury in comparison with cadet players. Nevertheless, no differences were observed between both senior and juvenile categories. This could be due to the fact that the number of cadet players injured was very low in comparison with those senior and juvenile athletes injured. 

The characteristics of the injuries observed in this study holds similarities with previous findings indicating a high prevalence of traumatic (acute) injuries compared with overuse ones, and a higher number of injuries sustained by back and wing players [20]. Further analysis indicated the absence of significant associations between the players’ position and the variables assessed. According to our data, the players’ category did not have a significant impact on the injuries sustained, except for the anatomical location. The OR indicated that senior players were more likely to sustain an upper-limb injury. We did not observe a high prevalence of shoulder injuries, as previously suggested by other authors [13]. Since participants in this study played at a regional level, a possible explanation for the low prevalence of shoulder injuries could be that the training load was not as high as to increase its occurrence [11]. In contrast, we did observe that the knee and the ankle were the anatomical locations most often injured, a finding that confirms the importance of developing injury prevention strategies focused on both articulations [4].

Finally, it is worth mentioning that most of the injuries reported in this study were either severe or moderate, and as a result, the average time absent from sport practice was around three weeks. These findings, coupled with the injury incidence observed, indicate that amateur female handball players are exposed to a substantial risk of sustaining an injury, even when playing at a regional level.

In addition, an interesting result of our research, is the fact that young players could not actively take part in physical education lessons for an average of 5 sessions, due to handball injuries. These findings are in line with previous observations indicating that as more children are becoming involved in organized sports practice, the incidence of injuries may have a detrimental effect on their participation in other healthy activities [27]. Studies performed with adolescents in the school setting also indicated that the handball practice was related with injury risk [28]. Our findings indicate that sport injuries not only affect the health of young players, but it might have a negative impact on adherence to physical education lessons.

Despite being one of the very few studies that provides information regarding sport injuries registered in amateur senior, juvenile and cadet intermediate level female handball players on one hand, and also analyses the influence of the players’ category on injury incidence on the other, some limitations should be acknowledged. For instance, we only included two teams per category, thus the sample size was somehow small. In addition to this, coaches failed to report injuries sustained during training and during competition separately, a fact that made it impossible to make direct comparisons with other studies regarding injury incidence. Lastly, we were not able to analyze the influence of training characteristics (i.e., internal and external load), or the maturation level on injury incidence. 

## 5. Conclusions

In a small sample of senior and young female handball athletes, who played at and amateur and intermediate level, a substantial risk of sustaining a moderate or severe injury was observed. The players’ category did not seem to play an important role on the characteristics of the sustained injuries, except for the anatomical location. Regarding the possibility of being injured, our results indicated that cadet players showed a lower risk of sustaining a sport injury due to handball practice, in comparison with juvenile and senior players. 

## Figures and Tables

**Table 1 medicina-57-00565-t001:** Players’ characteristics and total number of injuries registered.

Number of Players	Senior *n* = 34	Juvenile *n* = 33	Cadet *n* = 47	Total *n* = 114
Age (years)	21.76 ± 2.34	17.23 ± 0.49	15.04 ± 0.48	18.01 ± 1.1
Height (cm)	168 ± 0.04	167 ± 0.03	164 ± 0.04	166.33 ± 0.03
Weight (kg)	66.86 ± 4.57	63.86 ± 5.28	59.5 ± 4.75	64.40 ± 4.86
Age Range				
Under-18		12(7.02%)		
Under-17		21(18.42%)		
Under-16			22(19,30%)	
Under-15			25(21.93%)	
Experience years	8.79 ± 3.55	7.5 ± 2.27	6.73 ± 1.48	7.67 ± 2.43
Player Position				
Goalkeeper	4(3.51%)	2(1.75%)	7(6.14%)	13(11.40%)
Center Back	7(6.14%)	5(4.39%)	8(7.02%)	20(17.54%)
Left/Right Back	8(7.02%)	11(9.65%)	13(11.40%)	32(28.07%)
Left/Right Wing	9(7.89%)	11(9.65%)	10(8.77%)	30(26.32%)
Pivot	6(5.26%)	4(3,51%)	9(7.89%)	19(16.67%)
Total of injuries registered				
Injuries registered by category	43(42.57%)	32(31.68%)	26(25.74%)	*n* = 101
Number of Players sustaining at least one injury	31(40.25%)	26(33.76%)	20(25.97%)	*n* = 77
Number of players who were not injured	3(8.11%)	7(18.91%)	27(72.97%)	*n* = 37

**Table 2 medicina-57-00565-t002:** Characteristics of the injuries registered in total and by category.

Variable	Category	Total Number of Injuries Registered *n* = 101	Total Number of Injuries in Senior *n* = 26	Total Number of Injuries in Juvenile *n* = 32	Total Number of Injuries in Cadet *n* = 43
Injury mechanism					
	Overuse	32(31.68%)	13(30.23%)	11(34.38%)	8(31%)
	Traumatic	69(68.31%)	30(69.77%)	21(65.63%)	18(69%)
Contact/No contact	Contact	53(52.47%)	20(46.51%)	18(56.25%)	15(58%)
	No contact	48(47.52%)	23(53.49%)	14(46.75%)	11(42%)
Type of injury					
	Sprain	31(30.69%)	13(30.23)	11(34.38%)	7(26.92%)
	Musculoskeletal	28(27.72%)	9(20.93%)	10(31.25%)	9(3.62%)
	Tendonitis	21(20.79%)	7(16.28%)	6(18.75%)	8(30.77%)
	Other	21(20.79%)	14(32.55%)	5(15.64%)	2(7.70%)
Type of Play Action	Change direction	58(57.42%)	23(53.49%)	19(57.38%)	16(61.54%)
	Jump/landing	27(26.73%)	13(30.23%)	9(28.13%)	5(19.23%)
	Throw	6(5.94%)	2(4.65%)	1(3.13%)	3(11.54%)
	Running	7(6.93%)	3(6.98%)	3(9.38%)	1(3.85%)
	Stopped	3(2.97%)	2(4.65%)	0(0%)	1(3.85%)
Playing surface					
	Cement/wood	71(70.29%)	34(79.06)	17(53.13%)	20(77%)
	Synthetic/wood	8(7.92%)		8(25%)	
	Synthetic	6(5.94%)			6(23%)
	Cement/synthetic	16(15.84%)	9(20.93%)	7(21.88%)	
Period of the season					
	Preseason	35(34.65%)	13(30.23)	15(46.88%)	7(26.92%)
	Season (1st round)	34(33.66%)	17(39.53%)	9(28.13%)	8(30.77%)
	Season (2nd round)	32(31.68%)	13(30.23)	8(25%)	11(42.31%)
Recurrent Injury					
	No	82(81.18%)	31(72.09%)	30(96.88%)	21(80.77%)
	Yes	19(18.81)	12(27.91)	2(3.22%)	5(19.23%)
Severity					
	Minor	2(1.98%)	1(2.33%)	1(3.12%)	0(0%)
	Moderate	53(52.47%)	18(41.86%)	19(59.37%)	16(61.54%)
	Major	46(45.54%)	24(55.81%)	12(3.50%)	10(38.46%)
Absent time					
	Weeks	3.52 ± 1.12	3.67 ± 1.30	3.66 ± 0.99	3.24 ± 1.09
	Training sessions	11.03 ± 4.16	12.81 ± 4.67	10.60 ± 3.42	9.68 ± 4.39
	Competitions	2.67 ± 0.87	2.93 ± 1.07	2.46 ± 0.73	2.64 ± 0.81
	Physical education sessions	3.93 ± 1.24		5.83 ± 1.86	5.96 ± 1.88

**Table 3 medicina-57-00565-t003:** Anatomical location of the registered injuries.

Location	Body Region	Total Injuries Registered = 101	% of the Total Injuries Registered	Total Injuries Registered in Senior *n* = 26	Total Injuries Registered in Juvenile *n* = 32	Total Injuries Registered in Cadet *n* = 43
Lower limb		51	50.50%	17(39.53%)	16(50%)	18(69.23%)
	Hip/pelvis	2	1.98%	1(2.33%)	1(3.13%)	0(0%)
	Knee	14	13.86%	5(11.63%)	4(12.50%)	5(19.23%)
	Ankle	19	18.81%	7(16.28%)	5(15.63%)	7(26.92%)
	Thigh	6	5.94%	1(2.33%)	1(3.13%)	4(15.38%)
	Leg	11	10.89%	3(6.98%)	5(15.63%)	2(8.33%)
	Feet/toes	0	0.00%	0(0%)	0(0%)	0(0%)
Upper limb		48	47.52%	26(60.47%)	15(46.88%)	7(26.92%)
	Shoulder	9	8.91%	8(18.60%)	0(0%)	1(3.85%)
	Elbow	7	6.93%	2(4.65%)	5(15.63%)	0(0%)
	Wrist	10	9.90%	4(9.30%)	4(12.50%)	2(7.69%)
	Arm	3	2.97%	2(4.65%)	0(0%)	1(3.85%)
	Forearm	4	3.96%	2(4.65%)	1(3.13%)	1(3.85%)
	Hand/finger	15	14.85%	8(18.60%)	5(15.63%)	2(7.69%)
Trunk		2	1.98%	0(0%)	1(3.13%)	1(3.85%)
	Breastbone/ribs	2	1.98%	0(0%)	1(3.13%)	1(3.85%)

**Table 4 medicina-57-00565-t004:** Exposure time per categories in female handball players (*n* = 114).

	Senior *n* = 34	Juvenile *n* = 33	Cadet *n* = 43
Average training length (minutes)	109.88 ± 7.02	117 ± 6	90
Training exposure (h)	5152	5110	5712
Number of Matches	38	40	76
Total Exposure (h)	5418	5390	6244

**Table 5 medicina-57-00565-t005:** Risk ratio of the relationship between category of female handball players (senior vs. cadet/juvenile vs. cadet) and risk of injury.

	Value	95% Confidence Interval	
		Lower	Upper
Odds Ratio for category (senior/cadet)	13.950	3.732	52.149
For cohort Injury	2.143	1.513	3.035
For cohort no injury	0.154	0.51	0.465
N of valid cases	81		
		Lower	Upper
Odds Ratio for category (juvenile/cadet)	6.075	2.111	17.749
For cohort Injury	1.923	1.329	2.781
For cohort no injury	0.316	0.147	0.680
N of valid cases	80		

## Data Availability

The data presented in this study are available on request from the corresponding author.

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
