# Peer review of "Sport Injuries among Amateur Women and Young Intermediate Level Female Handball Players: A Preliminary Investigation"

_medicina, 2021, doi:10.3390/medicina57060565_

Round 1
Reviewer 1 Report
Dear Editors,
Thank you for the opportunity to prepare a review about the manuscript titled: “Injury Pattern in Amateur Women and Young Intermediate Level Female Handball Players: Influence of the Sport Category”.
The Topic and idea of the article is interesting, original and actual, however, in my opinion some elements should be improved to increase it to a more useful level for readers and especially sports' environment. Article is prepared in very general style and in numerous places mental shortcut what give lack of information do not let to understand deeper sense of presented data analysis because reader have no chance to receive some key data to appropriate understand Author's way of thinking, e.g. there is lack of the characteristics of players (Subjects of the observation) in Material and Methods; lack of description of training and competitive system in Spanish handball on different level of experience.
According to the reviewer obligation I will present some more detailed points, below:
- Title:
I wonder about how to prove the “…Influence of the Sport Category” on the injuries. We have no data about training and competitive loads as well as other analysis to have an opportunity to fulfill it. Topic is very interesting but in the presented form and set of information it is impossible to be real to realize it.
On the other hand "Injury Pattern in Amateur Women and Young Intermediate Level Female Handball Players" suggested a wider data set. And in the present situation it is limited to the only relatively small representative of female handball players in all Spain. Authors should remember that describe group "only" 114 players in 3 groups.
My suggestion is to improve the title to be more informative and agree with content or suggest it is "preliminary investigation".
- Introduction
It is a very general set of information. In my opinion, Authors should add more details here connected with the title and substantiate the aim of the study.
a) it is a place where the definition of injury as key point to understanding the criteria in the article should be presented - Authors present it below (lines 85 - 89), however better sense is to show it in the introduction.
b) readers receive no information about characteristics of the training system in Spain, here or in the Material and Methods. It should be added.
c) the aim has two sub-points - first is clear and second "A second objective is to identify the influence of the sport category on the nature, extent and severity of the injuries observed" it is impossible to find answer without more detailed information about training system and differences in the internal and external loads among observed groups. Please discuss it once more. From my perspective it is general characteristics of injuries on different levels of training.
3. Materials and Methods
Should be a specific manual to help readers understand every step of the research designing process and be sure it is appropriate to use it to compare or receive information needed to repeat it in own research.
A. Participants
Authors should describe more detailed participants in every group as numbers, M+-SD for age, body height and weight, sport's experience (how long they play handball or whether participate also in other sports/training), positions of play, etc. with division for next category.
What was weekly training / competitive loads (how many training and matches that participated in general) and what capacity did they realize during the observed season? Did different groups realize this same work capacity? Were they participated only in the training and/or maybe also in PE lessons at school? Did they have any recovery system / biological regeneration in the club / home? It must be clear especially because Authors calculated for 1000h injuries' index. Did Authors check the level of maturation and use it in the analysis in younger groups?
Readers, however, have the opportunity to find some fragments of such information but they are scattered and very often confusing thanks to their different versions– here is the most important place to present everything clear and keep it in all manuscripts. Please improve it.
B. Data collection
Did Authors take into consideration differences between basketball and handball? or was the questionnaire standardized to any sports?
Line 82 - 83 "empty sentence" - "Information related to age, weight, height, date of injury, time spent per week on practice, time spent on games, field position, and how the injury occurred was also collected." - Authors did it but forgot to present it. It should be presented in the part of Participants, because it is basic information for analysis and an open opportunity to compare this results with other research. Please add it and use it!
Lines 91 – 96 – Authors know what is important, collected data but forgot to present this data or use it in detail during analysis. It should be added and used.
Lines 102 – 104 – RR index interesting, however here and in the part results need to be better and more clear describe, especially as number of samples as well as connections during calculations.
Line 106 – there is “confidence interval (IC 95%).” and should be … (Cl 95%)
- Results
Lines 109 – 110 – Authors present information about Participants which should be presented in the part Material and Methods (M&M) also with other characteristics.
line 116 - Authors present only here some information about playing positions and no words about it in the M&M or in other analysis. It would be interesting to add it to this article. Please think about it and try to use it.
line 121 – Table 1 – please re-write the title of the table and prepare a more clear version especially about numbers and their sense.
The numbers presented in table 1 are difficult to understand without clear description in the M&M and in the title of table. What is 100%? If we observe the total number of players 114 why we have presented numbers of players as Seniors n=43; Juvenile n=32 and cadets n=26 it is 101 similar to Injuries but not to Injured people or not injured.
Such a presentation is confusing. Please present the total number of players 114 divided for Seniors n=... Juveniles n= ... and Cadets n=... and every sum should be clear to understand the number of injuries, injured players in every group and not injured players. And if you concentrate on 77 players with injuries show it as 100% for analysis and show it in the title. It must be clear without mental shortcut. Please correct it.
Lines 132 and next – Table 2 think about more clear title
Please check the results because in numerous places you mistake numbers Seniors with Cadets and vice versa (n=26 vs. n=43)
Please explain what numbers are under category players in every category or injuries in the category. it is difficult to understand. What should the reader understand as 100% - general number of players, injured players or general number of every single injury?
Lines 140 and next – similar to table 2 and 1
Lines 148 and next – Table 4
I am confusing what does Authors means because they were used so different numbers before that I have problem to keep their track of thinking. the sum of players presented in the lines 109 and 110 and in the tables for groups are different? What does "N of valid cases" mean here?
I try to comprehend what you presented in table 4, probably only significant analysis. It should be added to description as well as you should add more clear description here and in M&M in the sub-point Statistical Analysis.
- Discussion
Line 156 to 170 - basic question - did presented authors use this same procedures, like Authors in the reviewed article? Did Authors use their own questionnaire or was a different set of points which were prepared by mentioned in this paragraphs Authors?
It could be a key to compare how Authors in relation other authors divide injuries and the only conversion factor (1000h) is not enough. Please check it carefully.
Line 179 – “It should be taken into account that in the aforementioned investigations, the sample was made up of athletes who trained many more hours than our players. The….” – How are the readers able to compare or check Author's statement without information about the training system and other data presented in the M&M and taking into consideration data analysis? Classical mental shortcut.
Lines 183 – 184 – empty sentence: "Nevertheless, our data indicate the existence of a considerable injury risk for amateur senior female players" - of course, however, please support it by arguments.
Line 186 – “Injury incidence was higher in senior than in the other two categories” The key to understanding it could be differences in the program and aim of the next categories. Look at e.g. LTAD programs / ideas there are clearly presented systematically increasing internal and external training/competitive loads. On the other hand you did not use any information about typical training loads analysis or stage of maturation. It could be the next important element(-s) which would explain differences connected with next levels of development and injurie prevention. Try to discuss and take it into consideration now and / or next time.
Line 192 – there is “u-17” should be used “U17”
On the other hand, beach handball and other beach games have different characteristics of external and internal loads as well as condition than indoor games. And it is needed to remember and especially when we are looking for the best disciplines to compare.
Lines 196 - 198 "Regarding RR, we found that senior and juvenile players had a significant higher risk of sustaining and injury in comparison with cadet players. Nevertheless, no differences were observed between both categories." - these two sentences are in opposition - significantly higher but no differences? Please prepare a better and more clear version for this paragraph.
Line 203 - second place where Authors presented positions on the pitch - but it is surprising and difficult to understand why they omitted this important element in the data analysis.
Line 216 - how Authors explain sudden elements of PE lessons and how readers should understand it? How is the organized handball training system in Spain? Is it realized during PE lessons or after obligatory lessons during separate training sessions?
Are observed players play handball both in training and PE lessons all the time or different, general preparation is the main accent during PE and specially during training in the club? It has to be clear to understand the system and causes of injuries. Please clarify it in the introduction and/or in M&M. It could be a reason that too much one sport discipline increases risk of injuries "thanks" to overloads.
Lines 220 - 227 - Limitations are a very important point here. However there are much more limitations which should be taken into consideration in the next version of the article and in future work. e.g. in the future Authors should think to invite physiotherapists or physicians (if they cooperate with the teams) who would much better prepare data to use in research. Readers have no idea when players were injured during training or matches or in other situations, etc.
- Conclusions
Line 230 - "The players’ category does not have a significant impact on the injury rate and on the injury pattern, except for the anatomical location. Coaches and fitness professionals should be aware that senior and juvenile athletes are more likely to be injured in comparison with cadet players." General yes, however, Authors presented a small part of the problem and on the example of a small number of players to generalize such conclusions. It should be limited to an observed group of female handball players and nothing more. Consciousness is very important but generally the most important is the quality of training and keeping rules and using not only special forms of training to prevent injuries. Unfortunately readers did not receive any information about the training system which could help us to understand causes of injuries. On the other hand it is connected also with knowledge about differences in the training with female athletes than with males athletes.
Please meet my comments and suggestions as friendly and helpful. I would like to motivate Authors to attentively read and header work to improve the quality of manuscripts as well to be better in scientific skills to present in the best way personal ideas and results of research to others.
My recommendation based on general assessment is MAJOR REVISION and a stronger analytical strategy on the base of better prepared research protocol description in material and methods.
With kindly regards
Jan
Reviewer 2 Report
Consider that the authors should make the following changes:
1. In the introduction they should significantly improve the identification of handball with other previous epidemiological studies and analyze the antecedent and mediating factors in the occurrence of injuries. In addition, it would be advisable to review the injury models, for example
Brewer, (1994), Heil, (1993). Wiese-Bjornstal, Smith, Shaffer and Morrey (1998), Andersen and Williams, (1988), Williams and Andersen (1998), Olmedilla and García-Mas (2009).
2. They should identify the type of research design and reference it adequately. For example: Ato, Manuel; López, Juan J.; Benavente, Ana (2013). A classification system for research designs in psychology. Anales de Psicología, 29 (3), 1038-1059.
3. There should be a procedural section. In this section they should specify whether it has ethics committee approval. Was the study conducted in accordance with the Declaration of Helsinki (WMA 2000, Bošnjak 2001, Tyebkhan 2003), which establishes the fundamental ethical principles for research involving human subjects?it should appear in procedure and not in a note at the end of the publication.
4. In the results, the following issues should be considered: (a) Workloads should be considered for the analysis. (b) It must be demonstrated that the sample is linear. Nomal and homoscedastic.
5. The conclusions obtained should be modified according to the issues raised above, especially through the workloads.
6. They cannot establish a lesion pattern using a chi-square.
Round 2
Reviewer 1 Report
Dear Authors,
Thank you for your answers and comments to my review. I accept it.
I read a new version of your manuscript with pleasure. It will deliver to people important information as well as I would like to encourage you to continue this topic and broaden it by inviting other levels of handball players.
Good luck!
Jan
Author Response
We thank this reviewer for the provided suggestion that helped to improve the quality of the manuscript
Reviewer 2 Report
Dear Authors:
Thank you for addressing the issues raised.
However, it is necessary that you indicate, in your reply letter to the reviewer, on which page and line the changes have been introduced.
In relation to demonstrating the linearity, normality and homoscedasticity of the sample, and considering that you expect significant differences, you should demonstrate that these have an effect size due to the variables considered and not to the effects of the sample distribution. It is necessary that they demonstrate it.
Yours sincerely.
Author Response
Thanks for your commentaries. We have added line numbers to the main document and to the response to the reviewers document (please, see attached file).
We have also included an analysis of the distribution and homoscedasticity, Please, see lines 159-62;184-86.
